# Long-Term Dynamics of Land Use in the Romanian Plain—The Central Bărăgan, Romania

**Adriana Bianca Ovreiu, Iulian Andrei Bărsoianu \*, Constantin Nistor, Alexandru Nedelea and Laura Comănescu**

Department of Geomorphology, Faculty of Geography, University of Bucharest, 030014 Bucharest, Romania; Adriana.bianca.ovreiu@drd.unibuc.ro (A.B.O.); constantin@geo.unibuc.ro (C.N.); alexandru.nedelea@geo.unibuc.ro (A.N.); laura.comanescu@geo.unibuc.ro (L.C.)
\* Correspondence: andrei.barsoianu@unibuc.ro

**Abstract:** Changes in land use and agricultural landscapes are primarily the result of socio-political and economic changes. This research is based on the analysis of old maps, pertaining to different historical periods, aiming to capture the dynamics of the landscape in the Central Bărăgan Plain. The cartographic materials used have the Map of Southern Romania from 1864, the Firing Master Plans and the Corine Land Cover dataset (1990, 2000, 2018) as reference. For the case studies, these sources are supplemented by the orthophotoplan from 2012 and the remotely-sensed image acquired by Corona satellites in 1974. The results highlight the fact that between 1864 and 2018, the Central Bărăgan Plain recorded important changes in land resources. The most significant transformations took place in the period between 1864 and 1959, when 58% of the plain area was purposed for another use in that large areas of land were introduced in the agricultural use. If in 1864 the agricultural lands represented less than half (196.896 ha) of the Central Bărăgan area, the natural areas being dominant (201.473 ha), in the first half of the 20th century, they increased exponentially (361.674 ha), the natural vegetation occupying much smaller areas (28.481 ha) mainly along the Ialomița and Călmățui rivers and near lakes. This trend is also expressed by the values of the index of naturalness which shows a drastic decrease, from 51.7% in 1864, to 10.6% in 1916–1959. There is also an increase of the area cover by settlements which should be explained by the occurrence of new villages and the increase in size of the existing villages. Another result of the approach concerns the changes that occurred in the agricultural landscape in the middle, respectively at the end of the 20th century, which is a fact quantified by computing the index of land fragmentation. The dynamics of land use and the changes in the features of the agricultural landscape, as far as the analyzed area is concerned, are due to the agrarian reforms pertaining to different historical periods, as well as to the forced relocations during the communist regime. Therefore, the study aims to highlight changes during historical, socio-economic and political time in land use and to reduce the degree of naturalness of the territory at the same time.

**Keywords:** land use dynamics; index of naturalness; land fragmentation; historical maps; reforms; deportations; political regime; Central Bărăgan; Romania

## 1. Introduction

Although land use was considered a local environmental issue, its importance as a global problem is increasing. Changes in landscape structure with respect to forested areas, agricultural land, water and air resources, throughout the planet, are justified by the necessity to produce food, water and shelter for over seven billion people. Farmlands, grasslands and urban areas have extended at a global scale in the last few decades, while simultaneously, energy, water and fertilizer consumption has considerably increased, and biodiversity has suffered substantial loss [1,2].



Croplands and grasslands have become together the planet's greatest biomes, followed by forests, and occupying about 40% of land surface. Changes in land plowing have doubled wheat harvests, which have increased in the past 40 years by approximately two billion tons per year. This growth could be partially attributed to the global expansion of farmland by approximately 12%, and particularly to technology development which led to the "Green Revolution", including high yield crops, chemical fertilizers and pesticides, advancements in mechanization and irrigation techniques. During the past 40 years, there has been a significant increase related to the global use of fertilizers by approx. 700%, as well as a growth of irrigation networks being deployed on cropland, by approx. 70% [3].

Changes in structure and functionality of ecosystems could affect the quality of their delivered services. Alterations concerning agricultural landscape could be caused mainly by changes in land use management policies and socio-economic transformations (transition from centralized control to a free market economy). These factors could have a major impact upon the features of agroecosystems, but could also influence the quality and amount of production per hectare [4–7].

Europe features great landscape diversity, which is a result of long-term dynamics in land use management. These changes in agricultural practice left a continuous mark upon landscapes and environment, in many cases with irreversible effects. Since a lot of activities related to land use are absolutely essential for the development of humanity, however, some land use practices degrade ecosystems and services we depend on, therefore long-term planning of land management becomes imperative [8]. Land use planning and management decisions are usually made at a local or regional level. However, the European Commission has an important role to play in ensuring that all Member States take environmental concerns into account in their development plans and apply integrated land management practices. Therefore, humankind is confronted with the challenge of reducing the negative impact upon the environment with respect to land use, while maintaining the same degree of socio-economic benefit [9].

For Romanian geographical research, landscape dynamics analysis is of utmost importance and necessity, both from a theoretical perspective, but especially for practical reasons, considering the accelerated tendency for landscape anthropization of flatland territories, including the Bărăgan Plain. Agricultural activity, forest management, industrial development, administration policies regarding territorial planning influence landscape changes [10–12].

The Bărăgan Plain has suffered transformation from the earliest times, since the emergence of the first settlements, accompanied by fallows, deforestation and opening of great land areas to agriculture. The intensification of agricultural practice created a necessity for developing irrigation systems. Political changes from the middle and second half of the 20th century have also imposed intense transformation upon agricultural landscape, manifested through changes in the degree of landscape fragmentation [13,14]. The Bărăgan is dominated by rural and agricultural landscapes, and changes in agrarian practices or land ownership rights confirm the necessity of analyzing interactions between the anthropic factor and other components of the environment.

The main objectives of the present research reside in emphasizing the land use dynamics and changes of naturalness degree and agricultural fragmentation, in a politic and socio-economic different context which led to the adoption of agrarian reforms with the manifested aim of changing land ownership rights, as well as to repressive actions to support forced displacement of population from different regions of the country to the Bărăgan Plain. All these changes took place in a natural favorable context to agricultural activities.

Therefore, the current study presents the transformations which occurred over time in landscape structure and dynamics of the Central Bărăgan Plain, under the influence of the natural and anthropic factor.

## 2. Study Area

The Central Bărăgan Plain, also known as the Bărăgan of Ialomița, represents a sub-unit of the Bărăgan Plain, being presented in the geographic scientific literature as a most typical plain, of lacustrine or fluvio-lacustrine origin [15].

The Central Bărăgan is located in the south-eastern part of the country, in the eastern side of the Romanian Plain, overlapping the Ialomița–Călmățui interfluve (Figure 1). It is a transition between Southern Bărăgan (Mostiștea's Bărăgan) and Northern Bărăgan (Brăila's Bărăgan), as evidenced by both the geographical position and the lower thickness of the loess and the more argillaceous substratum. Considering the elevation, altitude decreases from northwest, where it reaches 90 m, towards the eastern side of the plain, where it has a value of 10 m, in the Ialomița's meadow [16–21].

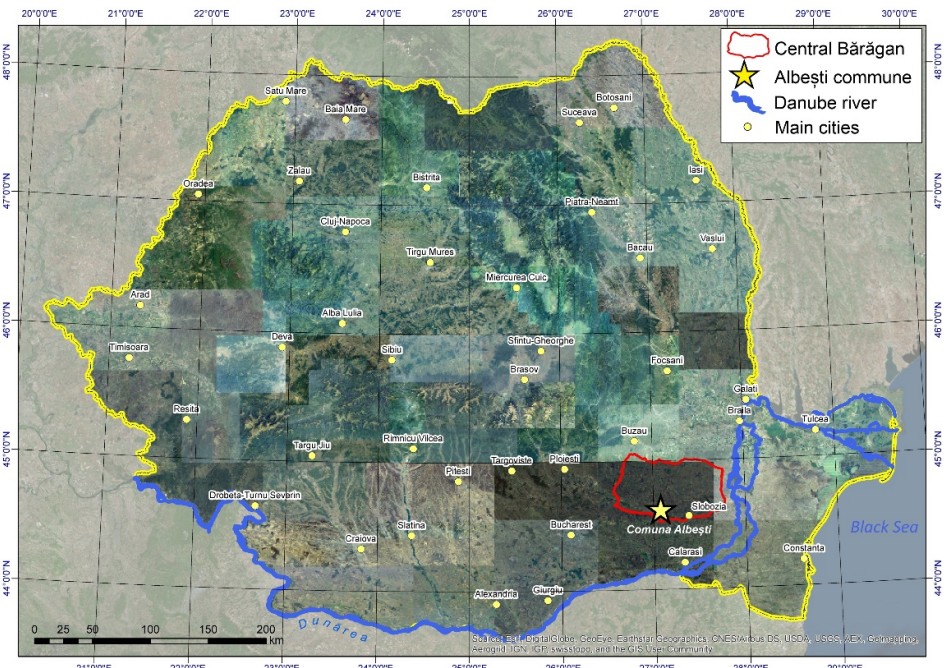

**Figure 1.** Central Bărăgan Plain—geographic location within Romania. Basemap provided by National Agency for Cadastre and Land Registration (ANCPI).

The Central Bărăgan Plain is characterized by a temperate continental climate, with extreme influences, with a high degree of continentalism, determined by the interaction of continental, arctic and polar air masses with maritime air, of oceanic or Mediterranean origin, as well as by relative homogenous properties specific to the structure of the active surface. Annual mean temperature features values lower than 10.5 °C in the western half and higher on the eastern side of the plain. Mean annual rainfall ranges between 450 and 550 mm, diminishing from west towards east and northeast, simultaneously with the increase of continentalism degree. The wind represents a climatic feature with great influence, most frequent in this region being the northern wind, followed by the northeastern and western ones [22].

From a pedological perspective, in the eastern half of the plain, where rainfall has slightly lower values, typical and calcic chernozem have developed, while in the western half, with slightly higher rainfall, cambic chernozems are present. Therefore, the zonal soils specific to this land unit belong to the Cernisoil class, being represented by the Kastanozioms and Chernozems [23].

The natural spontaneous vegetation from the Central Bărăgan Plain is specific to temperate grassland and forest stepped zone. Currently, areas covered with natural vegetation have mostly been fallowed and replaced by farmland.

## 3. Materials and Methods

The evaluation of the landscape dynamics in the Central Bărăgan Plain involved the use of old maps, pertaining to different historical periods. The analyzed cartographic materials include the Map of Southern Romania from 1864 (Szathmari), Firing Master Plans and the dataset known as Corine Land Cover (1990, 2000 and 2018). For the case studies, the orthophotoplans from 2012 and the Corona satellite images from 1974 are added (Figure 2).

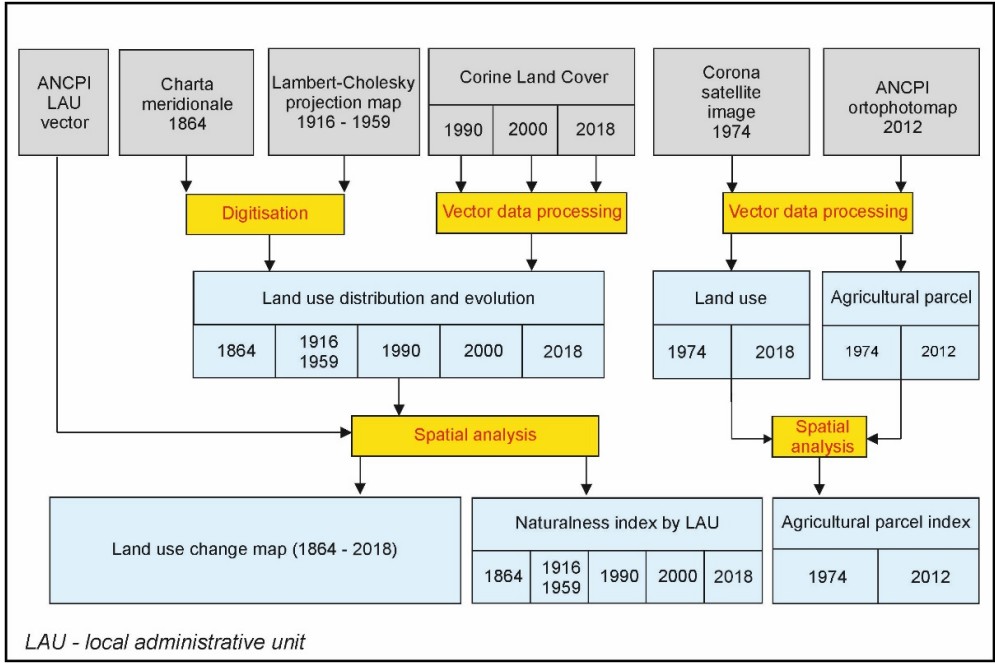

**Figure 2.** Methodological outline.

The Map of Southern Romania was created in 1864 and represents a detailed cartographic work which depicts Wallachia. It consists of 112 sheets and is also known by the name of its author, Szathmari [24].

In order to cover the entire territory of Central Bărăgan, we downloaded over 350 frames from the original map sheets. A direct download of the data was not possible, for which reason we needed to zoom in on interest areas (which resulted in images with a digital resolution of 1677 × 884 pixels), and then we chose the corresponding spatial reference system (Stereographic 1970) and the desired picture resolution (5 m/pixel). The frames, which were downloaded in GeoTIFF format, were mosaicked in ArcGis, using the Mosaic to New Raster function. Then, we limited the studied land unit, and the map made by mosaicking the downloaded sheets was resized according to it.

Afterwards, we began digitizing areas of different land use, for which we defined 5 categories: settlements, arable land, forests, water bodies, natural grasslands. Identifying these categories on the map was rather difficult, as the colors did not correspond to the legend, since the map has been deteriorated in the course of time.

We also downloaded 39 sheets of the Firing Master Plans [25]. Although the resulted mosaic is quite detailed, comprising more symbols and land use types, we used the same categories, established before with respect to The Map of Southern Romania from 1864. This approach was necessary in order to facilitate a subsequent comparison of these two maps, concerning land use.

For the assessment of changes and transformation of land cover in time we involved spatial analysis GIS-based techniques. For a comprehensive understanding of the land cover changes, we performed a statistical analysis for each category of class covers by

calculating of percent of changes and plot the correlation coefficient ($R^2$) between different data sources. The extraction of required data from the previously presented cartographic documents was done by means of manual digitization using specialized software (ArcGIS 10.2.2).

The distribution and dynamics of land resources for the year 1864 and for the first half of the 20th century was made using the Map of Southern Romania and military maps (Firing Master Plans). For the years 1990, 2000 and 2018, we employed the Corine Land Cover dataset [18–26]. The analysis was perform starting from the minimum mapping unit (MMU) of CORINE datasets which is 20 hectares and all the data extracted from historical and topographic maps after the geometrical operation were harmonized within this reference to make them able for computation. Based upon land use data specific for the five historical periods taken into consideration, we were able to create cartographic documents which could emphasize timespans when most significant changes of land use took place.

Using data from the National Agency for Cadaster and Land Registration (ANCPI) representing boundaries of local administrative units (LAUs), we computed the changes into index of naturalness (NI) for every period taken into consideration in the analysis. The NI computation was performed at the regional level of Central Bărăgan, as well as for each LAU (Figure 2). The NI was defined as the percent of natural or unartificialized surface from the analyzed area and measured the degree of naturalness of an ecosystem (In (%) = natural vegetation area/total plain area × 100) [27,28].

In order to emphasize the major influences of the socio-political factors with respect to the environment, we conducted a case study using a remotely-sensed image from 1974 [29,30] and an orthophotoplan from 2012 (ANCPI). Using these cartographic documents, we presented the transformations which occurred in land use and farmland distribution, 1974 being a representative year for the communist regime, while the year 2012 reflects the current situation. In order to quantify the degree of cropland fragmentation, we computed the index of land fragmentation [31] by dividing the total length of parcel boundaries by the perimeter of a rectangle of 1 km$^2$ in area. In this case, the higher the total length of the boundaries is, the higher land fragmentation would be (Figure 2).

## 4. Results

In 1864, Central Bărăgan was dominated by areas of natural vegetation, which occupied almost half of the plain's territory (48%). These were followed by croplands (47%) which were clustered in the proximity of settlements and along the most important water streams (Ialomița, Călmățui). Forested areas occupied 3% of the territory, which is over 12.616 ha, while settlements and water bodies accounted for 2% (5.652 and 2.907 ha, respectively) (Table 1, Figure 3).

**Table 1.** Area (ha) of lands with different uses on the Map of Southern Romania, Firing Master Plans and Corine Land Cover, in Central Bărăgan.

| Land Use (ha) | Map of Southern Romania (1864) | Firing Master Plans (1916–1959) | CLC (1990) | CLC (2000) | CLC (2018) |
|---|---|---|---|---|---|
| Cropland | 196.896 | 361.674 | 360.162 | 360.727 | 340.584 |
| Settlements | 5.652 | 12.985 | 24.930 | 24.711 | 24.865 |
| Forest | 12.616 | 12.265 | 12.894 | 13.496 | 14.662 |
| Water bodies | 2.907 | 4.130 | 8.527 | 8.282 | 4.945 |
| Grassland | 201.473 | 28.481 | 13.031 | 12.328 | 34.490 |

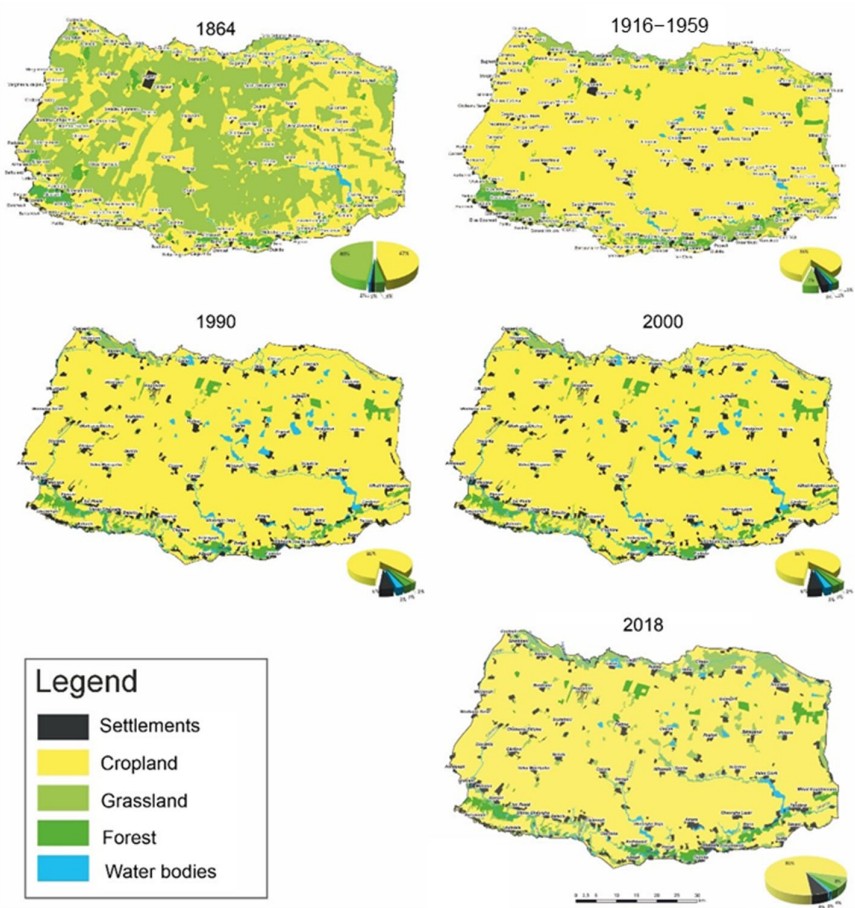

**Figure 3.** Distribution of land resources in Central Bărăgan, in 1864, between 1916–1959, 1990, 2000 and 2018.

Analyzing Figure 4 corresponding to the 1916–1959 historical period, we notice that in the first half of the 20th century, croplands were prevailing in Central Bărăgan (86%). Grasslands covered only 7% of the plain's territory, followed by forests and settlements which were represented by equal percentages (approximately 3%), while rivers and lakes accounted for 1% of the total area.

The superior fertility of chernozems and alluvial soils, the plain relief and climatic conditions represented favorable factors which contributed to the development of agricultural practice, manifested between 1864 and 1916–1959 through intense land exploitation and opening of great land areas to agriculture. This process harmed areas covered with natural vegetation, which were subjected to a drastic decrease. Although in 1864 agricultural land was present on almost half (196.896 ha) of the territory of Central Bărăgan and natural grasslands occupied a greater area (201.473 ha), the former witnessed an exponential increase (361.674 ha) in the first half of the 20th century, while natural grasslands covered a much lower amount of land (28.481 ha), mainly along the Ialomița and Călmățui rivers and in the proximity of lakes. This tendency for extending the cropland area is illustrated in Figure 4.

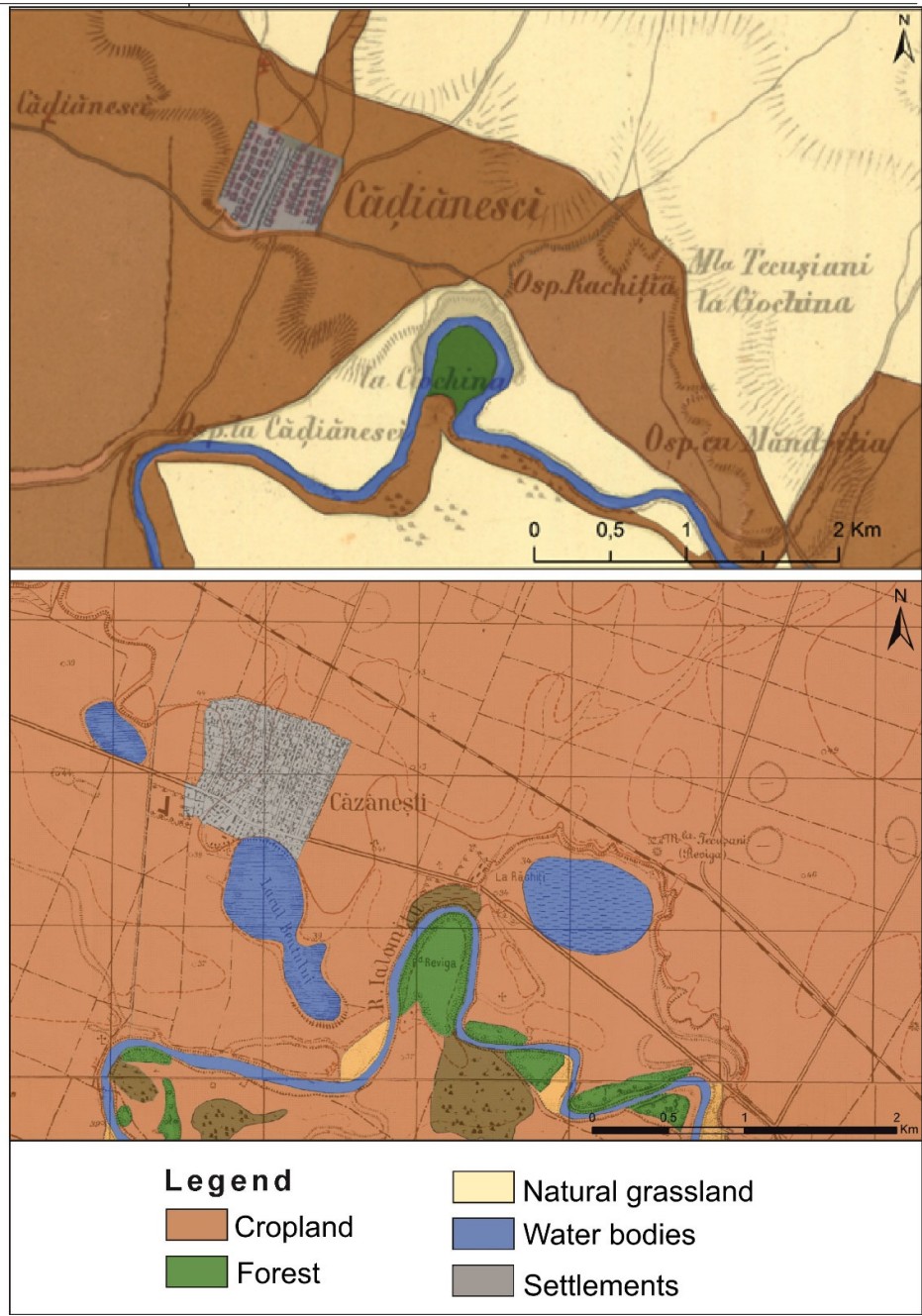

**Figure 4.** The extension of agricultural land in the proximity of Căzănești locality, in 1864 and the beginning of the 20th century.

The intensification of agricultural activity simultaneously raised the need for water resources, in order to supply the irrigation networks which operated during the dry seasons of the year. This factor justifies the doubling of areas consisting of water bodies between 1864 and the 1916–1959 period. Although, on the Map of Southern Romania (1864) water bodies occupied 2.907 ha, in the first half of the 20th century, their surface reaches 4.130 ha. As an example, we chose the Bent Lake, located near the locality of Căzănești (Figure 4) which did not exist at the time when the Map of Southern Romania was mapped in 1864. The purpose of this lake could be grasped from its very name (Bent in Romanian

language means "reservoir"); it was meant to be a reservoir, in order to supply water for nearby agricultural areas.

The territory occupied by settlements in the Central Bărăgan Plain indicates a value of 5.652 ha in 1864, which is 1% of the total surface of the study area. In the beginning of the 20th century, settlements were present on 12.985 ha (3%). The exponential growth of settled areas could be explained by the emergence of 60 additional localities, but also by the enlargement of the existing ones. There were also cases when localities which existed in 1864 are no longer present on the military maps (Firing Master Plans). We identified 20 such cases, caused whether by smaller settlements being included in the administrative territory of nearby localities which have spatially developed better, or because of their very small areas or population, which caused them to be disestablished later, although most of the populated areas have disappeared or have been displaced to the northern side of the plain, in the easily flooded meadow of the Călmățui river. Therefore, out of 150 localities observed on the Map of Southern Romania (1864), only 130 could still be identified after 1900, some of which changed their spatial extension and street network, but also their names. Consequently, the Firing Master Plans sum up 190 settlements.

For instance, the Smeiani de Josu locality was observed on the Map of Southern Romania in the meadow of the Călmățui river, in its branching area, being delimited by two river branches or secondary streams. The military maps though, show the fact that inside the administrative boundary of this settlement, there were areas with wet soils, characterized by excessive humidity and being supplied by multiple branches of the Călmățui river. Being an area with high risk of flooding, the settlers migrated to a nearby town, located to the south, on a terrace from the right side of the Călmățui river. The fact that this locality is not threatened by floods, granting optimal settlement conditions, could also be grasped from its name: Smeiani de Susu. Consequently, after the locality exposed to floods was moved, Smeiani de Susu did not only extend its administrative territory, but was also renamed as Smeeni.

An important reason which explains the growth of settlement-occupied areas in the Central Bărăgan Plain could be represented by forced delocations by means of repressive measures under the communist regime. Therefore, people from Banat and Oltenia, Bulgarians, Serbians, people from Basarabia, Germans, Jews etc., were cast out from their cities and were sent to other localities throughout the country or even in the middle of countryside, where they were forced to build their own villages. The most significant milestone of this process of forced population resettlement took place in June 1951, when 44,000 people from Banat and Oltenia were deported to Bărăgan [32]. There they established 18 new villages, of which only 4 are located in our study area: Fundata, Bumbăcari, Valea Călmățuiului and Mărculești (Table 2).

**Table 2.** Settlements established following mass deportations to the Central Bărăgan Plain.

| No. | Old Name | New Name | District | Region | Actual Name |
|---|---|---|---|---|---|
| 1. | Mărculeştii Noi | Viişoara | Slobozia | Bucharest | Mărculeşti |
| 2. | Giurgenii Noi | Răchitoasa | Feteşti | Constanţa | Giurgeni |
| 3. | Roşeţii Noi | Olaru | Călăraşi | Bucharest | Roşeţi |
| 4. | Jegălia Nouă | Salcâmi | Feteşti | Constanţa | Jegălia |
| 5. | Dâlga Nouă | Dâlga | Lehliu | Bucharest | Dâlga |
| 6. | Petroiu Nou | Movila Gâldăului | Feteşti | Constanţa | Gâldău |
| 7. | Feteştii Noi | Valea Viilor | Feteşti | Constanţa | Vlaşca |
| 8. | Perieţii Noi | Fundata | Slobozia | Bucharest | Fundata |
| 9. | Dragalina Nouă | Dropia | Călăraşi | Bucharest | Dragalina |

| 10. | Borcea Nouă | Pelican | Călăraşi | Bucharest | Cuza-Vodă |
| 11. | Cacomeanca Nouă | Ezeru | Călăraşi | Bucharest | Gradiştea |
| 12. | Borduşanii Noi | Lăteşti | Feteşti | Constanţa | Borduşani |
| 13. | Urleasca Nouă | Măzăreni | Brăila | Galaţi | Urleasca |
| 14. | Vădeni Noi | Zagna | Brăila | Galaţi | Vădeni |
| 15. | Tătaru Nou | Bumbăcari | Călmăţui | Galaţi | Bumbăcari |
| 16. | Stăncuţa Nouă | Schei | Călmăţui | Galaţi | Stăncuţa |
| 17. | Borcea Nouă | Frumuşiţa | Galaţi | Galaţi | - |
| 18. | Însurăţei Noi | Valea Călmăţuiului | Călmăţui | Galaţi | Valea Călmaţuiului |

Mărculeşti locality, from the county of Ialomita, has greatly extended following the arrival of deported people coming to this region. In 1864, the locality was known under the name Marculesci and it was very small. Although on the Map of Southern Romania it had an area of 24.81 ha, on the Firing Master Plans, its surface doubled, reaching 48.95 ha.

These unusual dynamics of settlement areas from the Central Bărăgan Plain are depicted in the following scatterplot (Figure 5). The graph expresses a correlation between the area covered by built space in 1864 and during the 1916–1959 period. The low value of the coefficient R (0.22), as well as the distribution of points in relation to the regression line that has a higher density in the area of origin of the axes, demonstrates the low degree of association of the two variables.

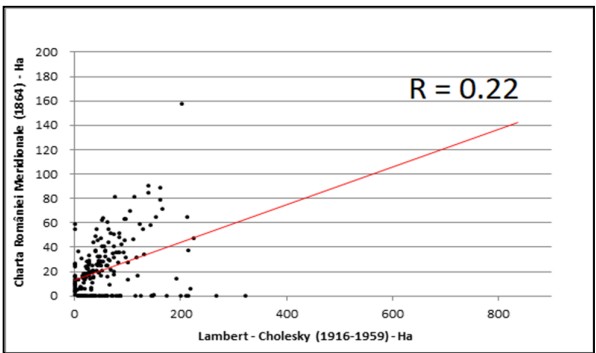

**Figure 5.** The progressive tendency specific to settlement areas in Central Bărăgan Plain.

Areas which suffered the smallest transformations correspond to forested landscapes, which occupied 12.616 ha in 1864, while in the first half of the 20th century, decreased by only 351 ha, reaching 12.265 ha. Such dynamics could be explained by the fact that the study area consists of a plain with natural vegetation such as grasslands and forested steppe, while forested areas only account for small areas, mostly clustered along the valleys of Ialomiţa and Călmăţui rivers. Being located in easily floodable areas, these small forests have not been exposed to intense anthropic intervention, because the respective areas were not considered suitable for being used as farmland or settling areas.

Allocation of land resources corresponding to 1990, 2000 and 2018 is displayed in Figure 3. In 1990, 86% of the Central Bărăgan was occupied by farmland with agricultural use. In contrast to the previous historical time frame (the beginning of the 20th century), we notice that the surface of settled space has doubled (6%), through expansion of built-up areas and emergence of new localities. Natural grasslands account only for 2% of the study area, having diminished by 5% when compared to the instance presented by Firing Master Plans. The forested areas and water bodies are represented in equal proportions, covering 3% of the study area.

Analysis conducted for the year 2000 has revealed only small land use transformations, which took place in the last decade of the 20th century, as well as an insignificant change in the spatial dynamics of land resources. Relatively higher modifications were made in the following period, corresponding to the year 2018, when cropland area diminished by 5% (81%), while natural grassland reached 9% of the study area, which is a significant increase compared to the previous two-time frames which were analyzed. Changes took place mainly in the eastern part of the Călmățui valley, in the easily-floodable meadow of the stream. The vulnerability of this area to floods has led to abandonment of certain farmland plots, followed by their renaturation with grassland.

Based upon land use maps for the analyzed timespans, we managed to compute the index of naturalness. Its highest values were recorded in 1864, reaching 51.7%. Following the spatial expansion of farmland and settlements at the expense of areas covered with natural vegetation, there was a drastic decrease in the values of this specific index, which dropped to 10.6% in the following period. The downward trend of the index of naturalness was maintained until the year 2000, when it reached its lowest value of 8.07%. A slight increase in the naturalness degree was observed in 2018, being triggered by a 5% decrease in cropland area and the simultaneous expansion of natural grassland, which occupied 9% of the study area during this year. The transformation took place particularly because of the abandonment of a few farmland plots located in the easily-floodable meadow of the Călmățui river, followed by their renaturation with grassland (Figure 6).

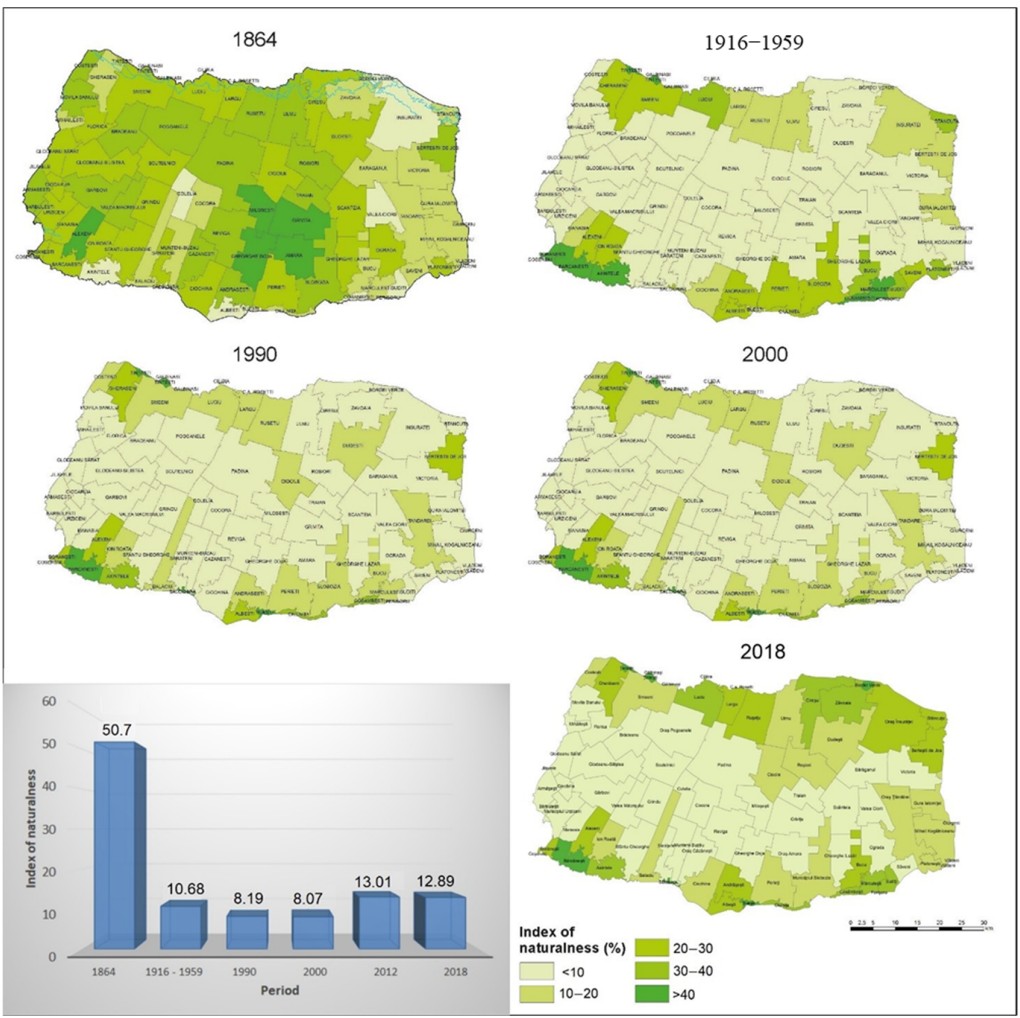

**Figure 6.** Index of naturalness for the years 1864, 1916–1959, 1990, 2000, 2018, in Central Bărăgan.

## 5. Discussion

Therefore, the Central Bărăgan Plain has suffered major changes between 1864 and 2018 in terms of allocation of land resources. The progressive tendency of areas corresponding to different land use categories is schematically represented in Figure 7, which leads us to conclude that there clearly is a relation of inverse proportionality between areas covered by grasslands and croplands. Although in 1864 the ratio between these two land use categories was quite balanced, in the following period, we notice a strong growth of farmland areas, simultaneously with a radical decrease in natural vegetation regions until the middle of the 20th century, while in the following years, they exhibited relatively stable dynamics, characterized only by small variations. A similar tendency is specific to settled areas, which recorded a significant increase until the end of the last century, reaching a maximum surface in 1990. The following period featured extremely low variation in the dynamics of settled areas. Water bodies displayed an important increase in the second half of the 20th century, when the intensification of agricultural activity raised the need for creating water reservoirs, in order to supply irrigation networks. In 1990, the area of water bodies reached its maximum value at 8.527 ha. In the following decade, this value remained rather constant, being diminished afterwards until 2018, when it dropped to 5.956 ha. The diminish of water surface is due to removal of same agriculture lands which led to the abandonment of artificial water stores used for irrigation. This situation is coupled with the severe drought register between 2010 and 2012 in Romania as a consequence to Global Climate Change [33]. The slightest changes and dynamics were observed with respect to forested areas, which displayed variations close to 13.000 ha for the whole study period.

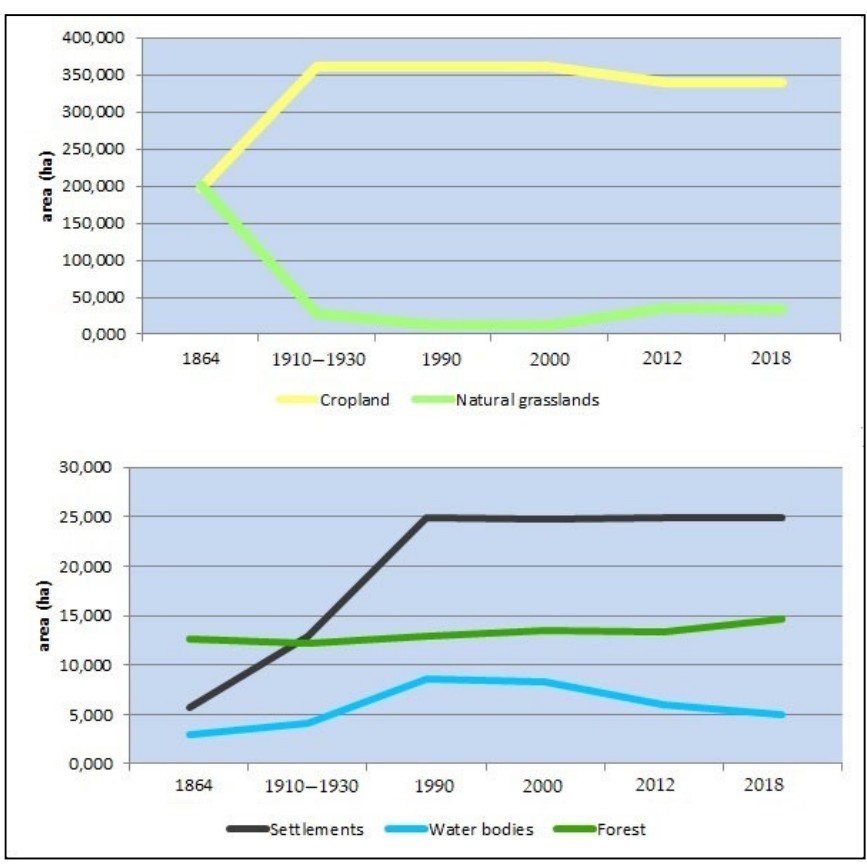

**Figure 7.** Land use dynamics in Central Bărăgan Plain (1864–2018).

The most significant changes took place between 1864 and 1959, when 58% of the territory of Central Bărăgan changed its use. Wide areas were followed in order to be prepared for agricultural use. Areas which constantly maintained their specific destination represent 42% of the Central Bărăgan Plain, being used as farmland even since 1864 and until the present day. This time frame was followed by the 1959–1990 period, when major changes were recorded especially with respect to settled areas, which reached a much lower value (14%), when compared to the previous situation. Changes in land resources were also observed between 2000–2012 (11%), when croplands from the eastern half of the Călmățui valley were abandoned and regained their natural state. The most reduced dynamics of land use was recorded for the periods 1990–2000 and 2012–2018, representing a value of only 4 percent, respectively, 3 percent (Figures 8 and 9).

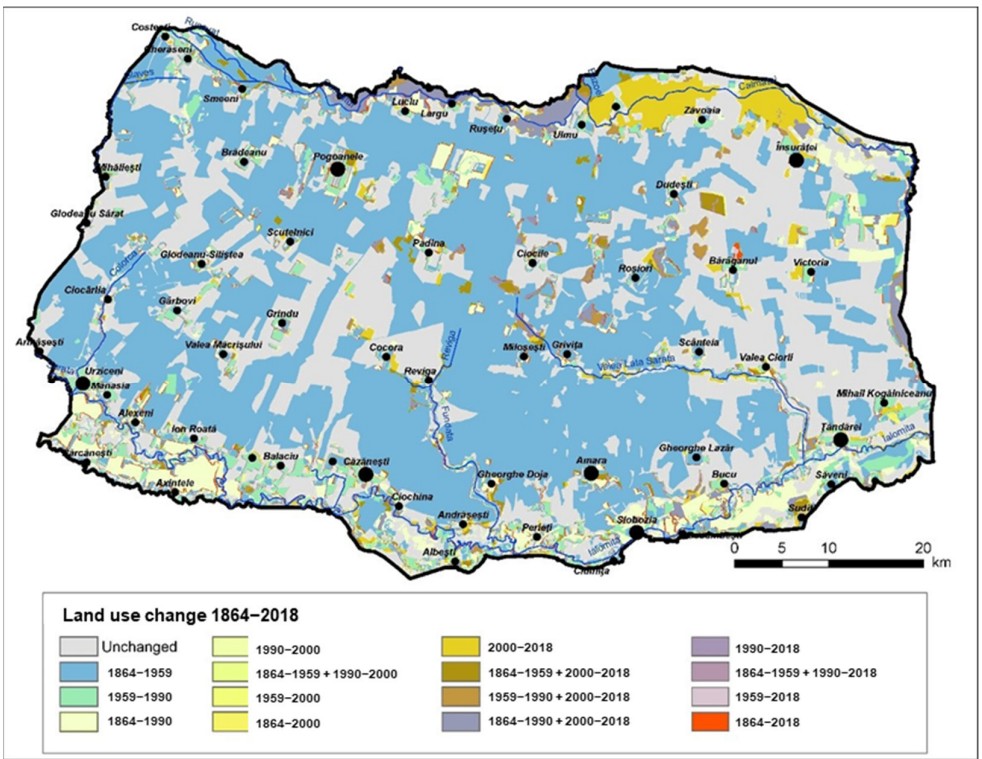

**Figure 8.** Surfaces affected by successive land use changes during the analysis period 1864–2018.

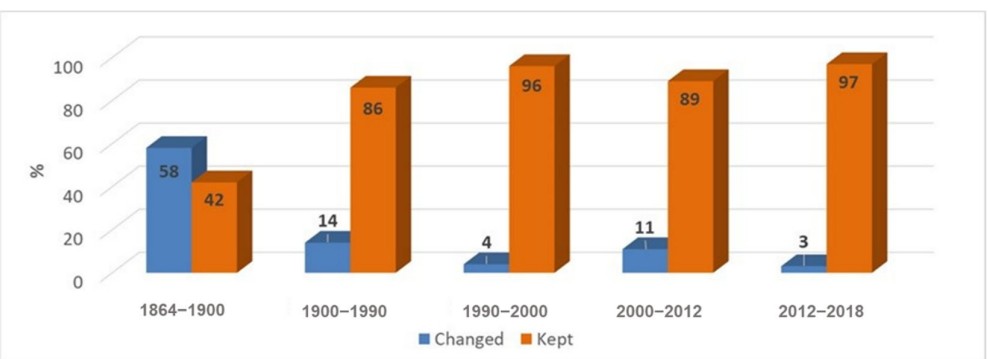

**Figure 9.** Surfaces of land which has acquired a different usage category (%) between 1864–2018 in Central Bărăgan.

Changes occurred in the landscape of LAUs, as well as in the aggregation or parceling degree of farmland will be exemplified through a case study of a local administrative

unit of the Central Bărăgan Plain. The main objective of this analysis is to illustrate the influence of socio-politic factors upon changes related to the local environment.

In order to observe transformations imposed in the landscape of Albești village, located in the southern extremity of the Central Bărăgan Plain, we chose as reference years: 1974, which is representative for the communist regime and 2018, to present the current situation.

As far as land use is concerned, differences between analyzed timespans are not very important. Both in 1974 and in 2012, we could observe a high percentage of land occupied by farmlands, which exceed in both situations an 80% of the total area of this administrative unit (Figure 10).

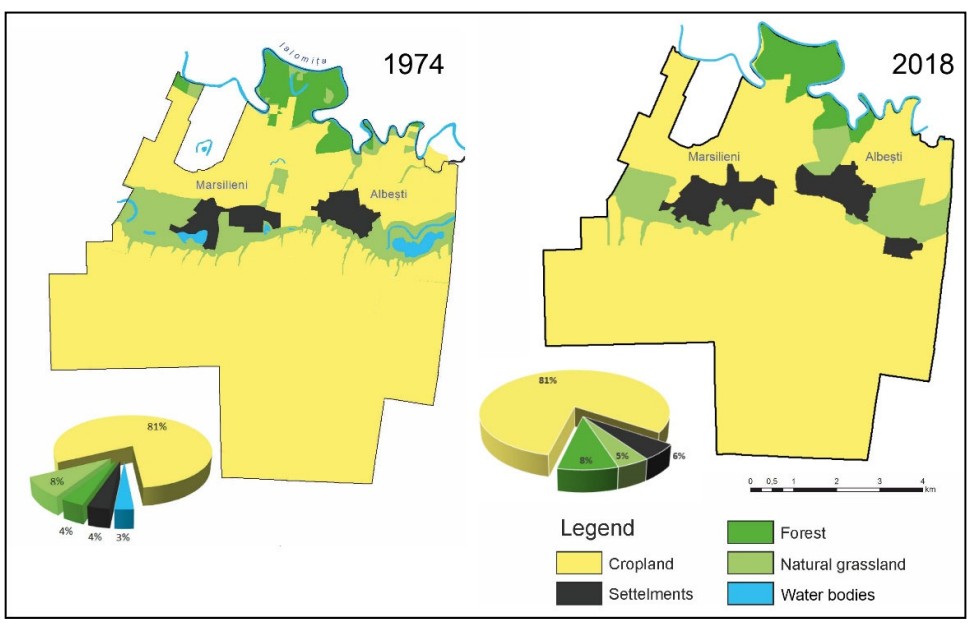

**Figure 10.** Spatial distribution of land resources in 1974 and 2018 in Albești village.

Political changes have affected land use policy across the whole country, including the arable landscape of Central Bărăgan Plain, displaying visible effects regarding especially the degree of land fragmentation of croplands.

In 1949, the process of agricultural collectivization began and consisted in the confiscation and aggregation of private croplands into agrarian farmlands administered by the state (collective farms). This action was based on the communist doctrine that small private farmland properties were inefficient, unprofitable and poorly developed from technological perspective. The solution to this problem was considered to be mass agricultural exploitation on extended areas, the state being the only entity which had the capability to efficiently administer these properties. Therefore, on one hand, the CAFs (Collective Agricultural Farms) were created (which were later changed into Agricultural Production Cooperatives—APCs), and on the other hand, the first State-owned Agricultural Farms (SAFs) were organized which consisted of land confiscated from great land owners and from previous royal domains. The State-owned Agricultural Farms have evolved into State-owned Agricultural Enterprises (SAEs), which numbered 419 entities in 1985 [34].

Plain regions represented the most suitable areas for implementing agricultural cooperative units, which is why croplands have been included in various APCs. Such an administration system has generated significant changes with respect to peasant's ownership rights upon their land, people losing their ability to make decisions related to exploitation of farmlands [17,35].

Therefore, the socialist agricultural system was characterized by aggregation of croplands, an extensive exploitation of great agricultural areas and a consistent homogeneity of different categories of land use, as it results from the analysis of farmland allotment map for the year 1974 (Figure 11).

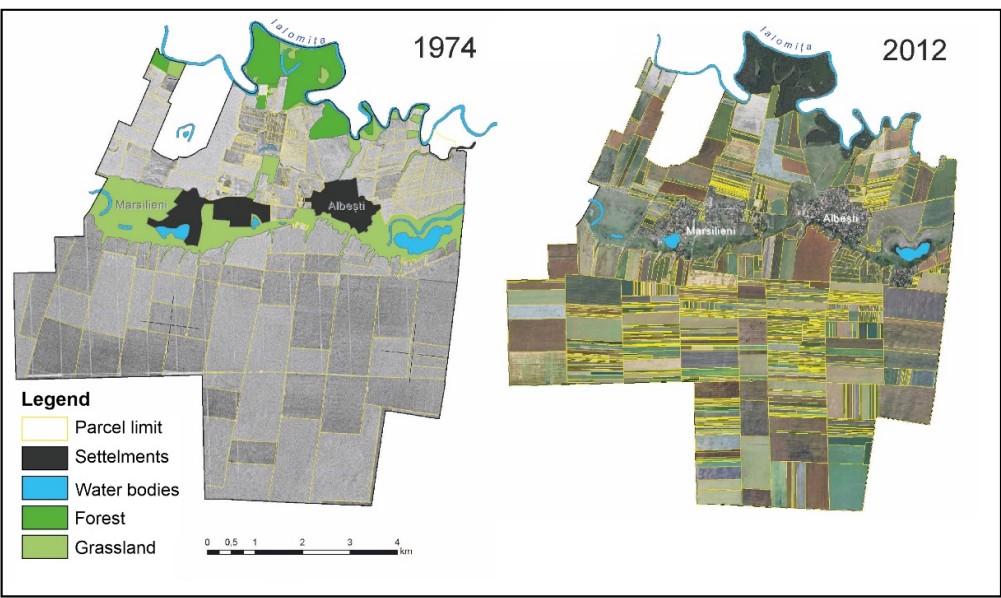

**Figure 11.** Land fragmentation in 1974 and 2012 (Albeşti village).

After 1990, the centralized land management system was discontinued and the APCs ceased to operate, all these being followed by mass expropriation of land and reenactment of private property in agriculture.

The conversion of Romanian agriculture after 1989 from the cooperative system to private property has led to major transformations in the agricultural landscape located in the case study area, mainly expressed through a higher degree of land fragmentation (Figure 11). This phenomenon of intense allotment of agricultural land is characteristic for the entire territory of Romania, since after 1990, the average size of farms decreased to 2.5 ha [36].

Property atomization and the predomination of small-sized agricultural exploitation is the consequence of legislative measures of expropriation "usually on the previous locations'' owned before the creation of APCs and represents a phenomenon which encumbers modern agricultural practice [37]. On the other hand, the greatest impact of the high degree land fragmentation of cropland is reflected upon the structure and functionality of agricultural landscape.

Crumbling of agrarian property had moral and social advantages (moral regain of the owner), but also economic disadvantages, concerning the low productivity of the new type of small private agricultural exploitation. In this context, the current tendency is of aggregating cropland (especially arable land) in agricultural associations, in order to increase the economic efficiency and profitability [20,35].

In order to quantify fragmentation of allotments, we computed the index of land fragmentation. The higher the value of the index, the higher land fragmentation would be. The socialist agricultural management system features much lower values of this index, compared to the decentralized and capitalist system of farmland administration. From the cartographic representation of the parceling index (Figure 12), we could observe the fact that in 1974, it had a maximum value of only 27 km/km$^2$, the greatest area of the studied LAU being dominated by the 3–9 km/km$^2$ interval. In 2012, the situation changed,

with the fragmentation index reaching much higher values, even of 80 km/km², most of the study area displaying values between 20–50 km/km².

Values of this index are relevant both for the landscape of Central Bărăgan Plain, as well as for the entire territory of our country. A high degree of cropland fragmentation is a current issue in post-communist Romania, which hinders modern agricultural practice at European standards.

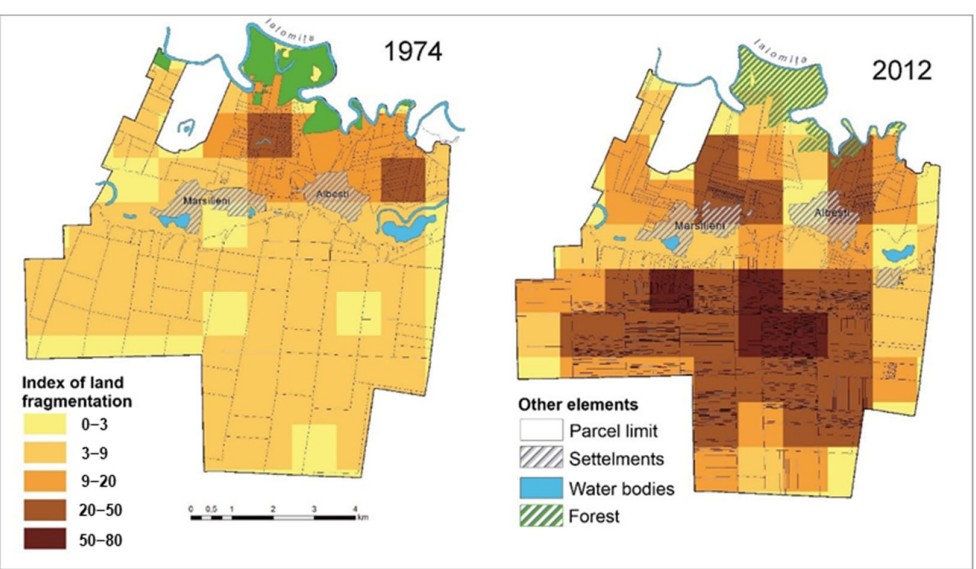

**Figure 12.** The index of land fragmentation in 1974 and 2012 (Albești village).

Therefore, landscape dynamics in Central Bărăgan Plain has been greatly influenced by the political regime which ruled the country before 1989, through agrarian reforms and policies adopted during this period of time, which have affected ownership rights and farmland aggregation degree. Political decisions which were made after this year, such as the return to private property and mass expropriation of land have increased the degree of farmland crumbling.

## 6. Conclusions

The Central Bărăgan Plain features a dominant agricultural landscape, historical transformation upon it being ordered by two major categories of factors: natural and anthropic.

The present research illustrates important changes related to land use management throughout different time frames, marked by political and economic events with great implications. The trend of human pressure upon landscape was also demonstrated, by means of computing the degree of environment artificialization.

Our research based on the integration of different cartographic data source manages to prove what happened with landscape from the Romania steppe in the 19th and 20th century, when natural grassland areas were considerably reduced in favor of croplands, and the index of naturalness dropped under 10%. The built-up areas of settled space display a continuous increase towards the end of the century, while after 1990, their dynamics were significantly reduced. The socialist system had directly impacted this element, through its forced deportation policy, which displaced large populations from other areas of the country and moved them to Bărăgan Plain.

As the results reveal the main driven force in the land cover/use changes was the political system beginning with the liberalization of the grain trade at the beginning of the 19th century and continuing with the land nationalization of the big farms at the end of

the First Word War and continued by communism system in the middle of the 20th century. Within the beginning of the 21th century, new rules based on the market economy produced new mutation in the land cover classes beginning with an increase of land fragmentation and ended with a unification of agriculture land into big farms nowadays.

One of the major limitations was the lack of accurate data for the same time period of investigation and this introduced an unequal distribution of our analysis over time with different time-lapse intervals. Other gaps were the uneven scales and legends that we have to harmonize and unify to provide affordable dates.

**Author Contributions:** For conceptualization, A.-B.O.; methodology, A.-B.O. and I.-A.B.; software, I.-A.B. and C.N.; validation, A.-B.O., L.C. and A.N.; formal analysis, A.-B.O., C.N., L.C.; investigation, A.-B.O. and A.N.; data curation, A.-B.O. and C.N.; writing—original draft preparation, A.-B.O.; writing—review and editing, A.-B.O., I.-A.B. and L.C. All authors have contributed equally to the development of this article. All authors have read and agreed to the published version of the manuscript.

**Funding:** This research received no external funding.

**Institutional Review Board Statement:** Not applicable.

**Informed Consent Statement:** Not applicable.

**Data Availability Statement:** The data presented in this study are available on request from the corresponding author.

**Acknowledgments:** When gathering data, we received the support of the Ialomița County Prefecture and Slobozia Town Hall, whom we thank.

**Conflicts of Interest:** The authors declare no conflict of interest.

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
