# Peer review of "Long-Term Dynamics of Land Use in the Romanian Plain—The Central Bărăgan, Romania"

_agriculture, doi:10.3390/agriculture11050423_

Round 1

Reviewer 1 Report

The graphic presentation of figures and tables could be improved, beside that I don't have any further comments.

Reviewer 2 Report

Greetings,
The authors followed the comments from the previous review. The paper is now accepted.

This manuscript is a resubmission of an earlier submission. The following is a list of the peer review reports and author responses from that submission.

Round 1

Reviewer 1 Report

The paper describes the dynamics and variability of land use changes since 1864 to present period at specific area of The Central Bărăgan plain in Romania. Authors have extracted digital land use data from various sources and used such data to describe the magnitude of land use changes and driving factors behind.

Regarding the manuscript, I have following comments:

Line 23: „There is also an increase in the number of localities and their area.“ It is not clear what authors mean. In case they refer to process of fragmentation, it is better to use the term number of patches and their size.

Line 151: „For the spatial analysis, change detection in time and the assessment of transformations related to land use in the study area, we have employed GIS-based techniques.“ Which techniques ?

Line 168: „Had we considered only the forested area, the results would have been irrelevant (Fig. 2).“. This sentence makes no sense to me.

Line 182: Different terminology was used in text (water bodies, rivers and lakes) and table 1 (wetland). Wetland should not be considered as same class as permanent water bodies. The terminology should be unified. If possible, the wetlands should be also mapped separetely, since this land use class is more sensitive to land use intensification.

Line 312: Which factors could possibly explain the drop of water bodies area in 2012?

Line 194: Table 1 should also include data from recent time periods such as 1990, 2000, 2012.

Line 330: Figure 7 is difficult to read. Does the colours represent different land use categories ?

Line 408: In the methodology section the authors described agricultural parceling index, in discussion section they reffered to index of parcel fragmentation. Terminology should be unified.

Other comments:

The title indicates that land use dynamics assessment was performed for the whole study area of the Plain, however some of the results refered only to selected case studies (parcel fragmentation). It would be

In the methodology section, the authors did not mentioned, how they dealed with the issue of minimum mapping units (spatial resolution), which usually differs between various historial periods and also between various data sources (hisorical maps vs. Corine data).

The data describing the „current“ situation are from 2012, which is already almost 10 years ago. However, Corine datasets are also available for period of 2018, thus I’m wondering, why authors haven’t used 2018 data.

Author Response

Response to Reviewers

Dear Reviewers thank you for making time to review our research, also we appreciate your valuable comments and suggestions.

Response to Reviewer 1

The paper describes the dynamics and variability of land use changes since 1864 to present period at specific area of The Central Bărăgan plain in Romania. Authors have extracted digital land use data from various sources and used such data to describe the magnitude of land use changes and driving factors behind.

Response to reviewer: Thank you for the understanding of our purpose and objectives

Regarding the manuscript, I have following comments:

Line 23: „There is also an increase in the number of localities and their area.“ It is not clear what authors mean. In case they refer to process of fragmentation, it is better to use the term number of patches and their size.

Response to reviewer: You have all right the sentence it is not clear; we reformulate it: “There is also an increase of the area cover by settlements which should be explain by the occurrence of new villages and the increase in size of the existing villages”

Line 151: „For the spatial analysis, change detection in time and the assessment of transformations related to land use in the study area, we have employed GIS-based techniques.“ Which techniques ?

Response to reviewer: We decided to follow you thought and reformulate all the sentence as: “For the assessment of changes and transformation of land cover and land use over time we have involved spatial analysis GIS –based techniques. For a comprehensive understanding of the land cover changes we perform a statistical analysis for each category of class covers by calculating of percent of changes and plot the correlation coefficient (R2) between different data sources.”  The term “change detection” was fired because it was used improper been related with remote sensing. 

Line 168: „Had we considered only the forested area, the results would have been irrelevant (Fig. 2).“. This sentence makes no sense to me.

Response to reviewer: We agree with you, after a briefly analysis we reformulate and improve the all paragraph.  “Using data from National Agency for Cadaster and Land Registration (ANCPI) representing boundaries of local administrative units (LAUs), we computed the changes into index of naturalness (NI) for every period taken into consideration in the analysis.  The NI computation was performed at the regional level of Central Bărăgan, as well as for each LAU (Fig.2). The NI was defined as the percent of natural or unartificialized surface from analyzed area and measure the degree of naturalness of an ecosystem (In (%) = natural vegetation area/ total plain acreage x 100) (Anderson, 1991; Machado, 2004).”

 Line 182: Different terminology was used in text (water bodies, rivers and lakes) and table 1 (wetland). Wetland should not be considered as same class as permanent water bodies. The terminology should be unified. If possible, the wetlands should be also mapped separetely, since this land use class is more sensitive to land use intensification.

Response to reviewer: We have considered your suggestion and unified the terminology, now the nomenclature used in Table 1 is identically with the charts and maps (Fig. 3, 4, 6, 10, 11, 12).

Line 312: Which factors could possibly explain the drop of water bodies area in 2012?

Response to reviewer: The removal of some agriculture lands has the consequence into the abandonment of artificial water stores used for irrigation. This was happened especially after 2012. A short note was also introduced in the main text.

“Water bodies displayed an important increase in the second half of the 20th century, when the intensification of agricultural activity raised the need for creating water reservoirs, in order to supply irrigation networks. In 1990, the acreage of water bodies reached its maximum value at 8.527 ha. In the following decade, this value remained rather constant, being diminished afterwards until 2012, when it dropped to 5.956 ha. The diminish of water surface is due to removal of same agriculture lands which led to the abandonment of artificial water stores used for irrigation. This situation is coupled with the severe drought register between 2010 and 2012 in Romania as a consequence of the Global Climate Change.”

Line 194: Table 1 should also include data from recent time periods such as 1990, 2000, 2012.

Response to reviewer: Your suggestion was considered and the table was completed with the information for the recent time : 1990, 2000, 2018

Line 330: Figure 7 is difficult to read. Does the colours represent different land use categories ?

Response to reviewer:  We agree with you and decided to improve the Figure 7 using different colors and symbols in order to make it more readable and easy to understand. The colors indicate the surfaces affected by successive changes into land use. Also we edited the caption of this figure. “Fig. 7. Surfaces affected by successive land use changes during the analysis period 1864-2018”

Line 408: In the methodology section the authors described agricultural parceling index, in discussion section they reffered to index of parcel fragmentation. Terminology should be unified.

Response to reviewer: Thank you for your comment.  The terminology was unified as index of land fragmentation

Other comments:

The title indicates that land use dynamics assessment was performed for the whole study area of the Plain, however some of the results refered only to selected case studies (parcel fragmentation). It would be

Response to reviewer: Yes, it is true, the land cover dynamics was performed for the whole plain within the help of CORINE datasets which are homogeneous.  The analysis of land fragmentation was performed only for designated area where we introduced more data from different source and at different resolution that needed to be harmonized with the CORINE. This situation illustrated in the study cases despite to be particularly are almost generally for all the study area.

In the methodology section, the authors did not mentioned, how they dealed with the issue of minimum mapping units (spatial resolution), which usually differs between various historial periods and also between various data sources (hisorical maps vs. Corine data).

Response to reviewer:  Thank you for the question and suggestion, we complete now the methodology: “The analysis was perform starting from the minimum mapping unit (MMU) of CORINE datasets which is 20 hectares and all the data extracted from historical and topographic maps after the geometrical operation where harmonized within this reference to make them able for computation”

The data describing the „current“ situation are from 2012, which is already almost 10 years ago. However, Corine datasets are also available for period of 2018, thus I’m wondering, why authors haven’t used 2018 data.

Response to reviewer: Thank you for this comment, you have all right the data must be updated. We decided to complete the methodology and introduce in our analysis the Coordination Information of Environment (CORINE) dataset for 2018. This reveal new and interesting situation that we underline in the main text. Moreover, this suggestion increases the value of our research and make it more actually.

Reviewer 2 Report

Greetings,
The paper needs to be corrected as follows:
In the introductory part the goal is too big. Reduce and add the scientific contributions of the paper. Add what answers this research should offer.
In the second part of the study area explain what was taken this year what about the new years because the last map from 2012 is 9 years ago. What is now the contribution of working with the last map which is 9 years old. What happened in the meantime. This needs to be stated in both the results and the discussion.
The results of Figure 5 show the period up to 1959, which is the contribution of this work, 62 years have passed. What happened in that period. Because this table is the only one that shows some results. All results should be in the selection of results and in the discussion these results should be explained in more detail. There are a lot of results in the discussion and few in the selection of results which should be the other way around. These two selections need to be corrected in detail. Increase the results, add some subselections and reduce the discussion.
In conclusion, provide guidelines for future research.
Strengthen non-Romanian references.

Author Response

Dear Reviewer thank you for making time to review our research, also we appreciate your valuable comments and suggestions.

Greetings,
The paper needs to be corrected as follows:

In the introductory part the goal is too big. Reduce and add the scientific contributions of the paper. Add what answers this research should offer.

Response to reviewer: Thank you for your comment! We decided to make the introduction more concise and summarize the objectives proposed. Also we connected our investigation to the mainstream flux of scientific research citing the authors contribution. Moreover, we defined the assumption that our research investigates.

In the second part of the study area explain what was taken this year what about the new years because the last map from 2012 is 9 years ago. What is now the contribution of working with the last map which is 9 years old. What happened in the meantime. This needs to be stated in both the results and the discussion.

Response to reviewer: Thank you for this comment. We complete the research with the CORRINE 2018 release that reveal new and interesting changes that we introduce in the results, discussion and conclusion sections.

The results of Figure 5 show the period up to 1959, which is the contribution of this work, 62 years have passed. What happened in that period. Because this table is the only one that shows some results.

Response to reviewer: This long period of 62 years with the lack of information was imposed by the missing of accurate cartographic source. For this interval we noticed an exponential increase of settlements surface due to the occurrence of new villages mostly formed by the deported of “unhealthy social class” between 1950 and 1960 which where abandoned in the steppe under the open sky and forced to build rudimentar shelters and produce food.

All results should be in the selection of results and in the discussion these results should be explained in more detail. There are a lot of results in the discussion and few in the selection of results which should be the other way around. These two selections need to be corrected in detail. Increase the results, add some subselections and reduce the discussion.

Response to reviewer: We considered you comment and reanalyzed the results and discussion sections, now the Results introduced all main find of this research and the Discussion analyzed in detail and evaluate this finds.

In conclusion, provide guidelines for future research.

Response to reviewer: Thank you for your comment. The Conclusion section was updated and completed with the main results of the research. Also we underline the degree of objectives completion and the limitation of this research.

Strengthen non-Romanian references.

Response to reviewer: The references were naturally completed due to the contribution of reviewers to improve our research.